# The Effects of Connectedness to Nature on Psychological Well-Being in Chinese Older Adults: A Moderated Mediation Model

**DOI:** 10.3390/bs15091200

**Published:** 2025-09-03

**Authors:** Lijun Zhu, Peimin Zhang, Song Gui, Chan Gao, Xiaofang Shen, Can Jiao

**Affiliations:** 1School of Psychology, Shenzhen University, Shenzhen 518052, China; 2050042007@email.szu.edu.cn (L.Z.); zhangpeimin2023@email.szu.edu.cn (P.Z.); 2200482037@email.szu.edu.cn (S.G.); gaochan@szu.edu.cn (C.G.); shenxiaofang2023@email.szu.edu.cn (X.S.); 2School of Government, Shenzhen University, Shenzhen 518052, China

**Keywords:** psychological well-being, connectedness to nature, self-perceptions of aging, meaning in life, moderated mediation, older adults

## Abstract

With the global aging population increasing, promoting psychological well-being (PWB) in later life has become critical. This study employed a cross-sectional design to examine the impact of connectedness to nature (CtN) on PWB among Chinese older adults, focusing on the mediating role of self-perceptions of aging (SPA) and the moderating effect of meaning in life (MIL). Data from 401 participants were analyzed using a moderated mediation model, controlling for demographic variables. The results indicate that CtN is positively associated with PWB, partially through reducing negative SPA. Moreover, MIL strengthens both the direct effect of CtN on PWB and its indirect effect via SPA. These findings underscore the importance of fostering nature connection and meaning in life to enhance well-being in older adults and offer valuable insights for interventions promoting healthy aging.

## 1. Introduction

China is undergoing a historically unprecedented demographic transition, with over 264 million people aged 60 and above now making up 18.7% of the population ([47]). This rapid aging, alongside accelerated urbanization and profound changes to traditional family and community structures, has introduced significant stressors for older adults, including rising social isolation, depression, and diminished quality of life ([14]; [67]; [72]; [77]; [81]). Large-scale research indicates that nearly one in four Chinese seniors experiences poor psychological well-being—a figure exacerbated by limited social safety nets and the erosion of traditional support systems, in contrast to many Western countries ([17]; [78]). As mental health challenges continue to mount, safeguarding and enhancing psychological well-being (PWB)—a multidimensional construct essential for both individual resilience and successful aging ([56]; [57])—has become a pressing public health priority. Unlike Western individualist paradigms that emphasize personal autonomy, Chinese elders’ well-being is embedded within a collectivist culture that values social harmony and interconnectedness ([15]; [29]). Therefore, advancing our understanding of psychological well-being in Chinese older adults is essential for informing culturally sensitive interventions and policies that support healthy and resilient aging amidst profound societal change.

In recent years, natural environments have been shown to enhance psychological well-being ([8]; [12]; [27]; [35]; [70]). For instance, the “20 min park effect” illustrates how brief, situational exposure to urban green spaces can improve mood and reduce stress ([79]). However, these state-like benefits are often short-lived, diminishing shortly after exposure ends ([32]; [46]). In contrast, connectedness to nature (CtN) represents a stable, trait-like psychological construct, reflecting an individual’s enduring emotional and cognitive bond with the natural world ([44]). According to Ecological Self Theory, individuals who perceive themselves as part of the natural environment experience enhanced psychological functioning ([7]). Similarly, Environmental Identity Theory posits that nature constitutes a core component of one’s identity, shaping attitudes and behaviors that foster well-being ([16]). Older adults, who often face barriers to frequent nature contact due to mobility or environmental constraints, require investigation into how this stable nature connectedness uniquely supports their mental health beyond the transient effects of brief exposure. Despite increasing evidence linking CtN to psychological well-being across populations ([31]; [53]), its specific role and mechanisms in older adults remain underexplored.

Self-perceptions of aging (SPA), defined as individuals’ cognitive and affective evaluations of their own aging process, are well-established predictors of psychological well-being ([4]; [40]). According to Cognitive Appraisal Theory ([38]), emotional well-being depends largely on these cognitive evaluations, with negative SPA linked to reduced life satisfaction and increased psychological distress in older adults ([60]; [69]). Although direct empirical evidence linking CtN and SPA remains scarce, theoretical perspectives suggest a plausible mechanism. Self-Perception Theory ([6]) posits individuals infer self-concept from behaviors and experiences. CtN may thus reduce negative SPA by fostering a stable, integrated self-view that includes nature as a valued identity component, alleviating feelings of vulnerability and decline associated with aging ([16]; [31]). This process supports older adults in reframing their aging experience and weakening adverse self-perceptions. Meta-analytic evidence further shows nature connectedness is positively associated with well-being, possibly through enhancing self-perceptions and reducing vulnerability ([11]). Accordingly, CtN maybe indirectly enhances PWB by diminishing negative SPA.

Meaning in life (MIL), defined as the perception that one’s life has purpose and significance, is a cornerstone of eudaimonic well-being ([65]). According to Self-Determination Theory ([19]), MIL reflects the satisfaction of basic psychological needs—autonomy, competence, and relatedness—and fosters adaptive cognitive and emotional functioning. Encompassing purpose, significance, and coherence, MIL enhances cognitive flexibility, emotional regulation, and self-integration, all of which are likely to promote more positive self-perceptions of aging and greater psychological well-being ([2]; [20]; [34]). Accordingly, MIL is posited to serve as a critical moderator in the present model, specifically moderating both the first stage of the mediation (CtN → SPA) and the direct path (CtN → PWB). For individuals with a strong sense of meaning, the restorative experiences of nature may be more readily internalized into a positive self-view, thereby strengthening the beneficial effect of CtN on SPA. At the same time, the sense of awe and transcendence evoked by nature may interact with a strong sense of purpose, resulting in a more pronounced direct enhancement of PWB.

Based on the theoretical and empirical foundations described above, this study proposes the following hypotheses:

**H1.** 
*CtN is positively associated with PWB among Chinese older adults.*


**H2.** 
*SPA may mediate the relationship between CtN and PWB.*


**H3.** 
*MIL may moderates both the effect of CtN on SPA and the direct effect of CtN on PWB.*


Notably, MIL is not hypothesized to moderate the second stage of the mediation (SPA → PWB), reflecting the strong, direct influence of internalized age perceptions on well-being as proposed by Stereotype Embodiment Theory ([39]). This theory suggests that while MIL can shape the formation of aging perceptions, it is less likely to buffer their direct negative effects.

To test these hypotheses, a moderated mediation model integrating cognitive and meaning-related factors is proposed (see Figure 1), aiming to elucidate mechanisms underlying aging and well-being and to inform culturally sensitive interventions promoting healthy aging.

## 2. Materials and Methods

### 2.1. Data and Participants

This study employed a convenience sampling method to recruit older adults from communities in Shenzhen, Guangdong Province, China. Participants completed paper-based questionnaires using one of two methods. In the first method, an experimenter conducted a one-on-one session with the participant. After explaining the purpose of the study and obtaining informed consent, the experimenter read each question aloud and recorded the participant’s responses. In the second method, participants read and completed the questionnaire independently, and the experimenter reviewed the completed form upon submission.

To ensure privacy, participants’ names were replaced with identification codes during data collection. Those who completed all questionnaire items received a small gift valued at approximately RMB 40 as a token of appreciation. The incentive protocol was approved by the Institutional Review Board (IRB) of Shenzhen University (PN-202200105).

The final analytic sample included 401 older adults, ranging in age from 60 to 91 years (Mage = 67.38, SD = 5.62), of whom 74.3% were female.

### 2.2. Measure

#### 2.2.1. Dependent Variable

Psychological well-being was assessed using the revised Chinese version of the Ryff Psychological Well-being Questionnaire ([55]), adapted by [74] ([74]). Wu revised the original 84-item scale, based on both English and Chinese versions provided by Professor Carol Ryff, to better align with the Chinese cultural context and accurately reflect psychological well-being among Chinese individuals. The adapted scale contains 20 items, making it more suitable for use with older adults. Response options ranged from 1 (“strongly disagree”) to 6 (“strongly agree”), with higher scores indicating greater psychological well-being. Cronbach’s alpha for this scale in the current study was 0.73.

#### 2.2.2. Predictors

Connectedness to Nature (CtN) was measured using a shortened version of the 20-item Disposition to Connect to Nature Scale (DCN), originally developed by [10] ([10]). This scale assesses specific behaviors that reflect an individual’s experiential closeness to the natural environment, relying on self-reported engagement with nature-related activities. Prior research ([49]) suggests that the DCN is particularly suitable for populations such as older adults and children, as it emphasizes concrete behavioral expressions of nature connectedness rather than abstract attitudes, enhancing ecological validity. Based on the item difficulty distribution reported by [10] ([10]), 20 items were selected to align with the behavioral patterns and cognitive characteristics of older Chinese adults, ensuring a comparable item difficulty range. Sample items include: “I talk to animals,” “I consciously watch or listen to birds,” and “My favorite place is in nature.” Rasch modeling followed [10]’s ([10]) specifications, using joint maximum likelihood estimation. All fit indices met standard criteria (mean square fit statistics [MNSQ] within the acceptable range of 0.7 to 1.3, PSR > 0.8). Detailed item content, parameter estimates, and psychometric properties are provided in the Appendix A. Cronbach’s alpha for this scale was 0.79.

Self-perceptions of aging (SPA) were assessed using the 17-item Brief Aging Perceptions Questionnaire (B-APQ; [63]). Participants rated each item on a 5-point Likert scale from 1 (“strongly disagree”) to 5 (“strongly agree”), with higher scores indicating more negative perceptions of one’s own aging. Cronbach’s alpha for this measure in the present study was 0.81.

Meaning in life (MIL) was measured using the “Presence of Meaning” subscale of the Meaning in Life Questionnaire ([66]). This subscale has been widely used in studies of older adults ([23]; [26]), emphasizing the role of experienced meaning in later life. It includes five items rated on a 7-point Likert scale ranging from 1 (“absolutely untrue”) to 7 (“absolutely true”). Sample items include “I understand my life’s meaning” and “My life has a clear sense of purpose.” The total score was calculated by summing the five items. The subscale demonstrated good reliability in this study (Cronbach’s alpha = 0.78).

#### 2.2.3. Control Variables

In line with previous studies ([21]; [51]; [67]), the analysis included three control variables: Gender, Age, and Self-reported health. Preliminary tests showed that other demographic variables, such as education and income, were not significantly associated with the key study outcomes and were thus excluded from the final analysis due to their non-significance.

### 2.3. Analysis Plan

Data analyses were conducted using SPSS (version 22.6) and the PROCESS macro for SPSS. To assess the presence of common method bias, Harman’s single-factor test was performed, revealing no significant bias in the data ([52]).

The analysis proceeded in three stages. First, descriptive statistics and Pearson correlations among the key variables were computed using SPSS (version 22.0) to provide an initial overview of the data. Second, the mediation hypothesis was tested using Model 4 of the PROCESS macro, examining the indirect effect of CtN on PWB through SPA while controlling for age, sex, and self-reported health. Bootstrapping with 5000 resamples was conducted to generate 95% bias-corrected confidence intervals (CIs), with indirect effects deemed significant if the CIs did not include zero. Third, a moderated mediation analysis was conducted using Model 8 of the PROCESS macro to determine whether the indirect effect of CtN on PWB via SPA varied as a function of MIL. Conditional indirect effects were estimated using ordinary least squares regression with bootstrapped CIs at different levels of the moderator.

## 3. Results

### 3.1. Correlation Analyses

As shown in Table 1, CtN, SPA, and MIL were significantly and positively correlated with PWB; CtN, MIL and PWB were significantly and negatively correlated with SPA. Additionally, gender was significantly correlated with PWB, age was significantly correlated with SPA, while self-reported health was significantly correlated with SPA, PWB and MIL. Therefore, gender, age and self-reported health were included as control variables in the subsequent model analyses.

### 3.2. Mediating Effect Test

SPSS PROCESS Model 4 was used to examine whether SPA mediated the relationship between CtN and PWB ([28]). After controlling for gender, age, and self-reported health, mediation analyses were conducted (Table 2). The results of the first step showed that CtN significantly and negatively predicted SPA (B = −0.15, *p* < 0.01). When the mediator was not included, CtN significantly and positively predicted PWB (B = 0.27, *p* < 0.001). After including SPA in the model, SPA significantly and negatively predicted PWB (B = −0.40, *p* < 0.001), and CtN remained a significant positive predictor of PWB (B = 0.21, *p* < 0.001), indicating a partial mediation effect.

To further test the indirect effect, the bias-corrected percentile bootstrap method was used (Table 3). The results showed that SPA significantly mediated the relationship between CtN and PWB. The indirect effect was 0.06 (SE = 0.02, 95% CI [0.02, 0.10]), accounting for 22.22% of the total effect, indicating that SPA played a partial mediating role in this relationship.

### 3.3. Moderating Effect Test

Model 8 in SPSS Process was used to examine the moderated effect of MIL between CtN and SPA ([28]). The results after adding control variables (encompassing gender, age and self-reported health) are shown in Table 4. The interaction term of CtN and MIL had a significant predictive effect on SPA (*B* = −0.12, *p* < 0.01). Meanwhile, The interaction term of CtN and MIL also had a significant predictive effect on PWB (*B* = 0.07, *p* < 0.05).

To further examine the moderating role of MIL, participants were categorized into high and low MIL groups based on one standard deviation above and below the mean. Simple slope analyses were then conducted at these two levels of MIL (see Figure 2 and Figure 3).

Figure 2 illustrates that the association between CtN and SPA differs by levels of MIL. Specifically, CtN was significantly negatively associated with SPA in the high MIL group (slope = −0.26, *t* = −3.40, *p* < 0.001), whereas this association was non-significant in the low MIL group. This pattern indicates that CtN is more strongly related to lower (i.e., more positive) aging self-perceptions among individuals with higher levels of MIL.

Figure 3 shows a significant positive relationship between CtN and PWB in the high MIL group (slope = 0.22, *t* = 3.66, *p* < 0.001), whereas this association was not statistically significant in the low MIL group. This indicates that the positive effect of CtN on PWB is more pronounced for older adults reporting higher MIL.

## 4. Discussion

### 4.1. The Impact of CtN on PWB of Elderly Population

This study investigated the association of CtN with PWB and its internal mechanism. The model results indicated that CtN was significantly and positively associated with PWB, thus supporting Hypothesis 1. This result is consistent with previous findings from both East Asian ([43]; [76]) and Western contexts ([37]; [80]), highlighting the potential benefits of nature connection for the older adults. These findings align with the Selection, Optimization, and Compensation (SOC) framework ([3]). According to this model, older adults proactively select meaningful goals, optimize their resources, and compensate for losses to maintain functioning and well-being ([22]). Within this framework, CtN can be viewed as a key psychosocial resource that supports the application of SOC strategies. For example, CtN may enable older adults to select nature-based activities that are congruent with their physical capabilities (e.g., engaging in mindful gardening rather than strenuous hiking), optimize their sensory and emotional experiences through interaction with natural environments (e.g., appreciating the scents and sounds of a garden to enhance mood), and compensate for physical or social limitations by fostering symbolic or vicarious connections with nature (e.g., caring for indoor plants or viewing natural scenes). The SOC framework not only offers a theoretical explanation for the psychological benefits of CtN but also serves as a valuable lens through which emerging interventions can be interpreted. In China, recent initiatives such as horticultural therapy—which engages older adults in multi-sensory nature-based activities that support emotional and cognitive functioning ([42]; [75])—and community-based social healthcare practices (e.g., Nature-based Social Prescription; [45]) reflect a growing recognition of nature’s role in promoting well-being in later life. These interventions offer practical applications of SOC principles, suggesting that enhancing older adults’ connectedness to nature may represent a culturally grounded strategy for supporting psychological health amid rapid social change.

### 4.2. The Mediating Effect of SPA

The findings further revealed an indirect association between increased CtN and higher levels of PWB, with reduced SPA acting as an explanatory variable in this link, thereby supporting Hypothesis 2. Cognitive Reframing Theory posits that individuals actively reinterpret their experiences, shaping emotional and behavioral outcomes ([5]). In the context of aging, SPA refers to individuals’ cognitive and affective evaluations of their own aging process, which significantly impact their psychological adjustment and well-being. CtN may support positive cognitive reframing by immersing older adults in the natural cycles of growth, decline, and renewal, thus enabling them to perceive aging as a meaningful and life-affirming experience rather than one of inevitable decline ([16]; [59]). This shift may help reduce internalized ageist stereotypes and foster more adaptive self-perceptions. Prior research also confirms that SPA is a robust predictor of psychological well-being in later life ([36]; [40]).

These insights suggest that the psychological benefits of nature are not solely direct but operate through internalized aging beliefs. As such, interventions aiming to enhance well-being should not only promote nature connectedness but also address aging-related cognitions. In China, emerging evidence supports the value of nature-based programs for older adults. For example, short-term forest therapy activities (e.g., in tea forest settings: [41]), national forest health initiatives ([33]; [82]), have demonstrated improvements in emotional well-being, cognitive function, and sleep. While primarily targeting psychological and physical outcomes, these programs may also contribute to healthier aging self-perceptions by fostering restorative experiences and reducing stress. Combined with cognitive reframing components, such interventions may more effectively support healthy aging by reshaping how individuals perceive and adapt to the aging process.

### 4.3. The Moderating Effect of MIL

The findings suggest that the association between CtN and SPA varies by levels of MIL, with a stronger negative relationship observed among individuals reporting higher MIL, thus supporting the first part of Hypothesis 3. This is consistent with Cognitive Reserve Theory, which posits that individuals with greater psychological resources—such as a strong sense of meaning—demonstrate enhanced cognitive flexibility and coping capacity ([68]). Older adults with higher MIL may be better equipped to cognitively reframe age-related experiences, resisting internalized age stereotypes and fostering more adaptive self-perceptions ([68]; [50]). Prior research supports this notion, showing that individuals with greater MIL exhibit increased resilience to cognitive stressors and may more effectively utilize nature engagement to support positive aging narratives ([1]; [48]).

Furthermore, the results indicate that MIL also strengthens the positive association between CtN and PWB, supporting the second part of Hypothesis 3. This finding aligns with eudaimonic well-being theories, which regard MIL as a foundational psychological asset that enhances the capacity to derive meaning, fulfillment, and purpose from life experiences ([58]). When MIL is high, the restorative experiences afforded by nature are more likely to be internalized as personally significant and emotionally enriching, leading to greater well-being ([24]; [62]). This pattern is empirically supported: [31] ([31]) found that the benefits of nature connection on well-being were amplified in individuals with higher MIL. Moreover, a strong sense of meaning facilitates emotional regulation and resilience, further enhancing the psychological benefits of nature exposure ([65]).

Taken together, these findings underscore that the psychological benefits of CtN are not uniform across individuals. Rather, MIL appears to serve as a psychological catalyst that enhances the internalization of nature connectedness into positive self-perceptions and greater well-being. These findings further underscore the value of integrating nature connectedness with existential meaning-making processes when designing interventions for older adults. Promoting both CtN and MIL may yield more enduring improvements in psychological functioning among older adults.

## 5. Limitations and Future Directions

Several limitations of this study warrant consideration. First, although the widely used Connectedness to Nature Scale (CNS; [44]) has demonstrated sound psychometric properties in Chinese young adult populations ([25]; [71]; [76]), our pilot testing with older adults revealed difficulties in interpreting its abstract items. As a result, we adopted the Disposition to Connect to Nature scale (DCN; [10]), which emphasizes behavioral engagement. However, this alternative may not adequately capture the emotional and cognitive facets of nature connectedness. Future research should prioritize developing culturally and age-appropriate tools that holistically assess this construct, potentially integrating qualitative methods and behavioral observations to enhance validity (e.g., [54]).

Second, the cross-sectional nature of this study limits any inference of causality among CtN, SPA, MIL, and PWB. Although the proposed pathway from CtN to SPA is conceptually grounded in Self-Perception Theory, direct empirical support remains limited. To better understand the temporal sequence and underlying mechanisms of these associations, future research should employ longitudinal and experimental designs ([9]).

Third, the sample was limited to a specific cultural and socioeconomic context in China, which may restrict generalizability. Cross-cultural replication and inclusion of diverse demographic groups are essential to validate findings globally (e.g., [53]). In addition, the study employed a convenience sampling method, which limits the generalizability of the findings. Future research should aim for more diverse samples to improve external validity.

Fourth, to assess common method bias, we conducted Harman’s single-factor test. Given its known limitations ([30]), we recommend that future research adopt more rigorous approaches, such as multi-informant designs ([13]; [18]).

Fifth, while emerging interventions like virtual nature experiences show promise in supporting psychological well-being among older adults ([61]), the underlying mechanisms—especially their impact on aging-related self-perceptions and life meaning—remain underexplored. Investigating the combined effects of real and virtual nature exposure could lead to innovative, accessible strategies for promoting mental health in aging populations.

Building on the present study’s findings regarding the pivotal roles of CtN, MIL, and SPA in promoting psychological well-being among older adults, one potential example of an integrative intervention involves combining horticultural therapy with reminiscence therapy to target the identified psychological pathways. Horticultural therapy engages older adults in multi-sensory natural experiences that enhance emotional and cognitive well-being ([73]; [75]), while reminiscence therapy fosters reflection on personal life histories, promoting meaning-making and positive identity reconstruction ([64]). Integrating these two modalities—by incorporating structured life review and memory-based activities into horticultural settings—may synergistically enhance both nature connectedness and life meaning. Such a combined intervention aligns with the mediating and moderating mechanisms identified in this study and could more effectively promote adaptive self-perceptions of aging and psychological well-being. Future research should empirically evaluate the feasibility and effectiveness of this approach, particularly among community-dwelling older adults.

## 6. Conclusions

This study provides valuable insights into the psychological pathways linking individuals’ bond with nature to well-being in later life. The findings underscore the synergy between connectedness to nature, self-perceptions of aging, and meaning in life as interconnected psychological resources. The mediating role of self-perceptions of aging and the moderating function of meaning in life suggest complex, synergistic pathways through which the benefits of a nature connection are psychologically internalized and amplified. These findings offer significant practical implications for designing interventions that support psychological well-being in later life. Rather than solely promoting nature-based activities, integrative approaches that combine interventions for enhancing nature connectedness (e.g., horticultural therapy) with meaning-centered practices (e.g., structured reminiscence or group life story sharing) may more effectively foster resilient and adaptive aging. Compared to culturally generic methods, such context-sensitive strategies—aligned with collectivist values and responsive to rapid urbanization—are likely to achieve greater sustainability, community resonance, and long-term impact.

## Figures and Tables

**Figure 1 behavsci-15-01200-f001:**
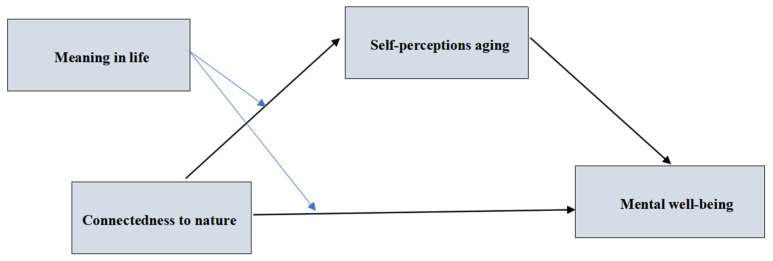
The proposed moderated mediation model to be tested.

**Figure 2 behavsci-15-01200-f002:**
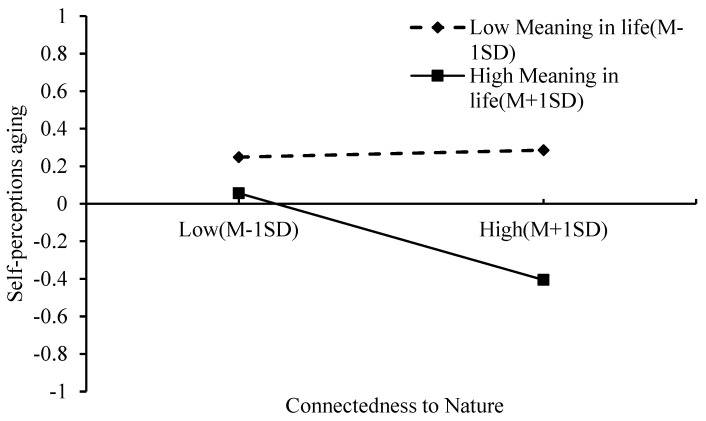
The effect of Connectedness to Nature and Meaning in life on Self-perceptions aging. Note: Self-perceptions aging values represent standardized scores; Connectedness to Nature and Meaning in Life are plotted at ±1 SD from the mean.

**Figure 3 behavsci-15-01200-f003:**
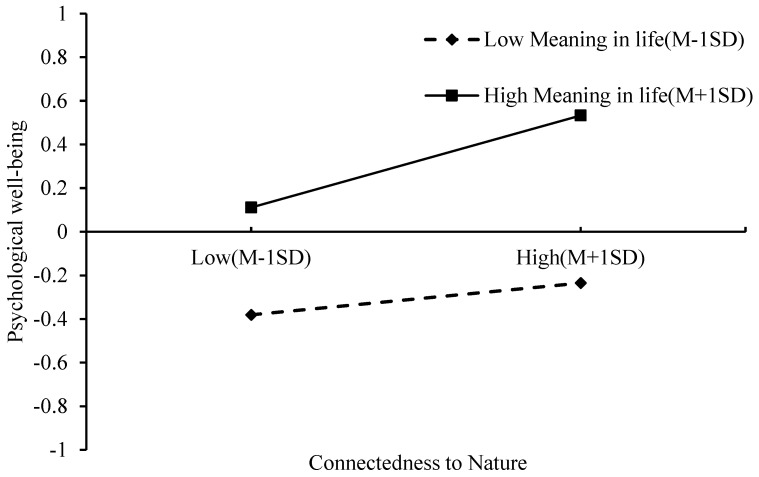
The effect of Connectedness to Nature and Meaning in life on Psychological well-being. Note: Psychological Well-being values represent standardized scores; Connectedness to Nature and Meaning in Life are plotted at ±1 SD from the mean.

**Table 1 behavsci-15-01200-t001:** Descriptive statistics and correlation analyses.

	1	2	3	4	5	6	7
1. Gender	1						
2. Age	−0.15 **	1					
3. Self-reported health	−0.10 *	−0.10 *	1				
4. Connectedness to Nature	0.03	−0.12 *	0.00	1			
5. Self-perceptions aging	−0.04	0.18 ***	−0.23 ***	−0.17 ***	1		
6. Psychological well-being	0.16 **	0.01	0.13 **	0.29 ***	−0.46 ***	1	
7. Meaning in life	0.05	−0.02	0.21 ***	0.31 ***	−0.28 ***	0.48 ***	1

Note: *N* = 401. * *p* < 0.05, ** *p* < 0.01, *** *p* < 0.001.

**Table 2 behavsci-15-01200-t002:** Mediated model test for Self-perceptions aging.

Regression Equation		Fitting Indicator		Coefficient Significance	
Outcome Variables	Predictor Variables	*R* ^2^	*F*	*B*	*t*
Self-perceptions aging	Gender	0.10	11.24	−0.08	−0.74
	Age			0.02	2.82 **
	Self-reported health			−0.32	−4.52 ***
	Connectedness to Nature			−0.15	−3.15 **
Psychological well-being	Gender	0.14	15.53	0.37	3.68 ***
	Age			0.01	1.78
	Self-reported health			0.22	3.34 ***
	Connectedness to Nature			0.27	6.28 ***
Psychological well-being	Gender	0.30	33.72	0.34	3.73 ***
	Age			0.02	3.30 **
	Self-reported health			0.09	1.49
	Connectedness to Nature			0.21	5.38 ***
	Self-perceptions aging			−0.40	−9.60 ***

Note: *N* = 401. ** *p* < 0.01, *** *p* < 0.001.

**Table 3 behavsci-15-01200-t003:** Decomposition of total effect, mediated effect and direct effect.

	Effect Value	Boot SE	Bootstrap 95% CI	Effect Ratio
Total Effect	0.27	0.04	[0.19, 0.36]	100%
Direct Effect	0.21	0.04	[0.04, 0.29]	77.78%
Indirect Effect	0.06	0.02	[0.02, 0.10]	22.22%

**Table 4 behavsci-15-01200-t004:** Moderating effect test of Meaning in life.

Regression Equation		Fitting Indicator		Coefficient Significance	
Outcome Variables	Predictor Variables	*R* ^2^	*F*	*B*	*t*
Self-perceptions aging	Gender	0.16	12.55 ***	−0.07	−0.64
	Age			0.03	3.57 ***
	Self-reported health			−0.25	−3.56 ***
	Connectedness to Nature			−0.11	−2.19 *
	Meaning in life			−0.22	−4.52 ***
	Connectedness to Nature × Meaning in life			−0.12	−2.99 **
Psychological well-being	Gender	0.40	36.71 ***	0.31	3.64 ***
	Age			0.02	2.46 *
	Self-reported health			0.01	0.20
	Connectedness to Nature			0.14	3.62 ***
	Self-perceptions aging			−0.33	−8.08 ***
	Meaning in life			0.31	7.82 ***
	Connectedness to Nature × Meaning in life			0.07	2.04 *

Note: *N* = 401. * *p* < 0.05, ** *p* < 0.01, *** *p* < 0.001.

## Data Availability

The data are not publicly available due to privacy restrictions, but they will be available from the corresponding author upon reasonable request.

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
