# Peer review of "The Effects of Connectedness to Nature on Psychological Well-Being in Chinese Older Adults: A Moderated Mediation Model"

_behavsci, 2025, doi:10.3390/bs15091200_

Round 1

Reviewer 1 Report

Comments and Suggestions for Authors

Summary: This manuscript investigates the relationship between connectedness to nature (CtN) and psychological well-being (PWB) among Chinese older adults, proposing a moderated mediation model where self-perceptions of aging (SPA) serve as a mediator and meaning in life (MIL) serves as a moderator. Using a cross-sectional survey with 400 participants and validated scales, the authors employ PROCESS analysis to test hypotheses. Findings suggest that CtN improves PWB both directly and indirectly by reducing negative SPA, and that MIL strengthens both pathways. It addresses a timely and important issue in the context of aging in place and community-based mental health interventions. The theoretical integration is strong and the moderated mediation model is appropriately chosen. However, the manuscript requires major revision due to issues with clarity, particularly English writing quality, as well as structure and interpretation of findings.

Strengths:

  • The topic is novel within the cultural context of China and aligns well with the journal's special issue on aging in place.
  • The theoretical rationale is sound, drawing from multiple psychological frameworks.
  • The statistical methods are appropriate and well-documented.
  • Measurement tools were thoughtfully adapted to an older Chinese population.

Weaknesses:

  • The writing quality is a major limitation with grammatical errors and awkward phrasing.
  • The introduction is overly long and could be more focused.
  • The hypotheses are not clearly and accessibly presented.
  • There is overstatement of causal conclusions in a cross-sectional design.
  • Practical implications are not deeply explored or connected to the findings.

Recommendation to reconsider after Major Revisions. The manuscript makes a meaningful contribution to the literature on psychological well-being in older adults, particularly within a Chinese context. However, the authors need to:

  • Revise the manuscript for language and clarity throughout
  • Reorganize the Introduction and Discussion for conciseness
  • Clarify and format hypotheses
  • Avoid overstating causal conclusions
  • Strengthen the practical implications and relevance to aging-in-place policies.

Line by Line

26 The inclusion of AI-based caregiving feels tangential and is not developed further. Consider removing or developing further.

34-36 PWB definition from Ryff is accurate. Consider briefly mentioning the cultural context for the construct's use in China.

45 Differentiate between CtN as a trait and transient exposure effects more clearly.

70-72 The link between CtN and SPA could be strengthened by citing studies that directly measure this relationship

57-104 Hypotheses should be numbered and visually separated for clarity. H1 H2 etc. Additionally, you could clarify that the moderation effect hypothesized applies only to stage one and the direct path.

117Convenience sampling can be acknowledged as a limitation (which it is later)

127 Clarify whether the incentive was IRB approved

174-177 You could consider if other covariates like education or income should be included

180 Harmans can be considered basic, you could mention this as a limitation and suggest future studies could use longitudinal or multi-informant methods

Figures 2 and 3 are helpful yet need clearer axis labels, scales, and captions.

249-64 Excellent use of the SOC framework. Consider noting whether these strategies are already common in China’s elder care practices.

286-291 The call for interventions is strong. You could suggest referencing existing community-based nature interventions in China or similar contexts

307-318 Reframe to emphasize associations, not causation

322-362 Excellent section, you are transparent about limits and provide thoughtful future directions

322-26 It’s great that you reflect on the limitations of the CNS scale. Consider stating whether the DCN version has been validated in Chinese populations before

370-73 practical implications here are strong, you could offer more specificity and reinforce the value of culturally situated interventions in your conclusion

Appendices/Psychometrics – strong reporting on psychometric adaptation & analysis, excellent transparency.

Comments on the Quality of English Language
  • The writing quality is a major limitation with grammatical errors and awkward phrasing. 
  • The introduction is overly long and could be more focused.
  • Revise the manuscript for language and clarity throughout
  • Reorganize the Introduction and Discussion for conciseness
  • Consider a round of professional editing

Reviewer 2 Report

Comments and Suggestions for Authors

The article submitted for review addresses a currently relevant topic concerning older adults. It utilizes a cross-sectional study design to examine the impact of connection with nature on well-being among older adults in China, examining the mediating role of self-perceived aging and the moderating influence of meaning in life. Almost all chapters are well-structured. The discussion raises some doubts. Two hypotheses were posed in the introduction, but they were neither refuted nor confirmed, which is essential.

Reviewer 3 Report

Comments and Suggestions for Authors

I wanted to start by saying that this study presents a well-articulated and theoretically exploration of an original moderated mediation model involving CtN, self-SPA, MIL as factors influencing PWB among older adults. There's no a lot of studies that touch on this, so I really appreciate the opportunity to review this manuscript. The manuscript honestly demonstrates a solid integration of theory and a rigorous methodological approach, making for an engaging and insightful read. This has certainly made my work a lot easier. I appreciated how the work advances my understanding by identifying the mediating and moderating processes that link CtN to PWB. It highlights the joint impact of cognitive (SPA) and existential (MIL) dimensions on well-being in older populations, which is crucial. The authors conducted the moderated mediation analyses using the PROCESS macro models, which was appropriate and added empirical rigor by controlling for important demographic variables, I would like to highlight that this helped to provide clarity about the conditional effects at play. I also appreciate the utilization of an adapted DCN scale specifically designed for older Chinese adults. The detailed adaptation process, covering translation, cognitive debriefing, and Rasch analysis, adds credibility and strengthens the construct validity of this measure in the cultural context being studied. However, I think it would be beneficial for the authors to discuss the potential impact of excluding emotional and cognitive components of CtN inherent in this behaviorally focused scale on the study’s outcomes. The findings have practical significance, suggesting that enhancing nature engagement and fostering a sense of meaning in life could effectively bolster well-being and self-perceptions of aging among older adults. The authors do a good job translating these findings into actionable intervention recommendations that integrate horticultural therapy with reminiscence therapy, which could serve as a promising approach to mental health promotion in later life. There are a couple of areas where further clarification or minor revisions could enhance the manuscript. Aside from what I mentioned above, and though the theoretical models are generally well-integrated, a more detailed explanation of why MIL moderates the CtN and SPA relationship but not the direct path between SPA and PWB would improve theoretical transparency would be ideal. There are also a few minor grammatical and typographical errors scattered throughout the manuscript that could be corrected to enhance readability, such as duplicated words and inconsistent formatting. So a round or two of proofreading would be helpful. I believe this manuscript makes a novel and meaningful contribution to the field and is suitable for publication pending these minor revisions.

Comments on the Quality of English Language

See comments to authors. 

Round 2

Reviewer 1 Report

Comments and Suggestions for Authors

Thank you for such a through and well explained revision!

Reviewer 2 Report

Comments and Suggestions for Authors

The authors took my comments into account. Thank you.